# A Comparative Study of Engineered Dermal Templates for Skin Wound Repair in a Mouse Model

**DOI:** 10.3390/ijms21124508

**Published:** 2020-06-25

**Authors:** Ilia Banakh, Perdita Cheshire, Mostafizur Rahman, Irena Carmichael, Premlatha Jagadeesan, Neil R. Cameron, Heather Cleland, Shiva Akbarzadeh

**Affiliations:** 1Skin Bioengineering Laboratory, Victorian Adult Burns Service, Alfred Health, 89 Commercial Road, Melbourne VIC 3004, Australia; ilia.banakh@monash.edu (I.B.); perdita.cheshire@monash.edu (P.C.); mostafiz.rahman@monash.edu (M.R.); h.cleland@alfred.org.au (H.C.); 2Department of Surgery, Monash University, 99 Commercial Road, Melbourne VIC 3004, Australia; 3Monash Micro Imaging, Monash University, 99 Commercial Road, Melbourne VIC 3004, Australia; Irena.Carmichael@baker.edu.au; 4Material Materials Science and Engineering, Monash University, 22 Alliance Lane, Clayton VIC 3800, Australia; premlatha.jagadeesan@monash.edu (P.J.); neil.cameron@monash.edu (N.R.C.)

**Keywords:** dermal templates, wound repair, human skin equivalent, graft, NovoSorb^®^ BTM, Integra^®^, inflammation, COX-2

## Abstract

Engineered dermal templates have revolutionised the repair and reconstruction of skin defects. Their interaction with the wound microenvironment and linked molecular mediators of wound repair is still not clear. This study investigated the wound bed and acellular “off the shelf” dermal template interaction in a mouse model. Full-thickness wounds in nude mice were grafted with allogenic skin, and either collagen-based or fully synthetic dermal templates. Changes in the wound bed showed significantly higher vascularisation and fibroblast infiltration in synthetic grafts when compared to collagen-based grafts (*P* ≤ 0.05). Greater tissue growth was associated with higher prostaglandin-endoperoxide synthase 2 (Ptgs2) RNA and cyclooxygenase-2 (COX-2) protein levels in fully synthetic grafts. Collagen-based grafts had higher levels of collagen III and matrix metallopeptidase 2. To compare the capacity to form a double layer skin substitute, both templates were seeded with human fibroblasts and keratinocytes (so-called human skin equivalent or HSE). Mice were grafted with HSEs to test permanent wound closure with no further treatment required. We found the synthetic dermal template to have a significantly greater capacity to support human epidermal cells. In conclusion, the synthetic template showed advantages over the collagen-based template in a short-term mouse model of wound repair.

## 1. Introduction

Large surface area wounds such as extensive deep burns can remain unhealed for many weeks and require intervention by skin grafting to accelerate repair and avoid chronic ulceration. Engineered dermal templates provide a scaffold niche where host cells can infiltrate to modulate the immune response and inflammation, thereby influencing wound repair. When dermal templates are applied and adhered to an open wound, it is clinically observed that uncontrolled granulation tissue does not develop, and the wound appears to be physiologically closed. The dermal templates’ physical (porosity, stiffness, roughness, degradation rate) and biochemical (receptor binding sites, toxicity, growth factors retention) properties ultimately influence the success of the graft. It is defined as early wound closure, timely vascularisation, minimal scar and wound contraction, and durable barrier function. Yet the key factors that influence these outcomes remain unclear.

In this study, we investigated the interaction between the wound bed and two “off the shelf” dermal templates in a full-thickness wound nude mouse model in order to define how the physical and biochemical properties of the scaffold can influence matrix-cell interaction and wound repair outcome. Both templates are bilayer constructs, with a superficial seal to exclude air from the healing wound surface. Once the deeper dermal template is vascularised from the wound bed, the upper seal must be removed and covered with epidermis (usually autologous split skin graft) for definitive healing. Integra^®^, and NovoSorb^®^ biodegradable temporising matrix (BTM) are the most common clinically used dermal templates for replacing skin defects [1,2,3]. Integra^®^ is a double layer material with an upper silicone film that simulates epidermis and a lower layer of porous (80 µm average pore size) bovine hide collagen and chondroitin-6-sulfate (a glycosaminoglycan or GAG from shark cartilage) cross-linked using glutaraldehyde. Whereas BTM is a fully synthetic polymer engineered from a biodegradable polyurethane with a temporary polyurethane seal, eliminating any potential risk of cross-species residual antigenicity or disease transmission, with visible pores of 188 ± 84 μm in size. Dermal grafts were compared to full-thickness skin allografts with respect to inflammation, proliferation and wound repair. Allografts and xenografts are well tolerated in nude mice due to their lack of mature T lymphocytes, and therefore, defective adaptive immune response. However, their innate immune response is still intact to generate inflammation [4].

The transition from inflammation to granulation/proliferation stage is a critical step to secure healing progress. This transition is tightly orchestrated with a range of growth factors and chemokines which originate from the bone marrow or the wound bed. Often it is the balance between these factors, rather than their presence or absence, that alters the wound microenvironment and defines the wound outcome. Interleukin 1 beta (IL-1β), interleukin 6 (IL-6) and tumour necrosis factor-alpha (TNF-α), and chemokine CC motif ligand 2 / monocyte chemoattractant protein-1 (CCL2 / MCP-1) are considered key proinflammatory growth factors, whereas interleukin 10 (IL-10), transforming growth factor-beta 1 (TGF-β), chemokine CC motif ligand 1 (CCL1 / TCA-3) are thought to act as anti-inflammatory and wound healing signals. A prolonged inflammatory phase can result in excessive ECM deposition by myofibroblasts and hence fibrosis, which leads to clinically unsatisfactory scarring outcomes [5,6].

The dermal templates’ mechanical forces can also influence wound healing by activating specific signal transduction pathways. It is thought that grafting dermal templates inhibits wound contraction, resulting in reduced scarring and fibrosis. Although this mechanism is not fully understood, its effect has been linked to inflammation response and fibroblast to myofibroblast transformation that occurs in a deep dermal skin injury [7,8,9,10].

Here, we studied animal-derived collagen-based and fully synthetic dermal grafts for wound contraction, inflammation, vascularisation, and host cell infiltration. Using molecular analysis, we explored mediators in control of these processes and identified significant differences in the matrix–cell interaction. Moreover, the dermal templates’ capacity of supporting a living bioengineered human skin equivalent was also studied here. This, to our knowledge, is the first study in which the interaction between the wound bed and dermal grafts has been explored. 

## 2. Results

### 2.1. The Synthetic Dermal Template with Larger Pore Size Vascularises Faster than Collagen-Derived Dermal Template 

Full-thickness wounds were grafted with synthetic BTM, Collagen-derived Integra, or allogenic native mouse skin. The grafts were analysed using H&E staining (Appendix A). Graft vascularisation, measured by endothelial cell marker CD31 expression, was used as an indicator of graft take two weeks post grafting (Figure 1a,c–e). The middle of the grafts was analysed separately from the edge of the graft as an indication of the endothelial cell infiltration from the wound bed. It is important to measure the infiltration of dermal templates from the wound bed in a mouse model, as the wound edge contribution in healing large surface area grafts in patients would be relatively small. The immunohistological analysis showed significantly more extensive vascularisation in BTM, compared to Integra^®^ grafts (***p* ≤ 0.01). In all grafts, CD31^+^ endothelial cells populated the wounds upwards and inwards from the wound bed and the wound edge, although BTM grafts were able to home CD31^+^ endothelial cells from the wound bed faster than Integra^®^. The wound bed contributed equally to vascularisation compared to wound edge in BTM grafts. Vessels were also larger in BTM compared to Integra (Figure 1d,e). In allogenic native skin grafts, an intermediate level vessel density was observed. This can be a mix of pre-existing graft vasculature and neovascularisation [11]. Figure 1b represents the scanning electron microscopy images of cross-sections of Integra^®^ and BTM compared to native skin highlighting their difference in pore size, shown in dark grey.

### 2.2. Host Dermal Fibroblast Infiltration Correlates with Vascularisation 

Fibroblast growth into dermal templates was detected by immunohistochemistry using vimentin specific antibody (Figure 2). Vimentin is a type III intermediate filament, widely used as a universal marker for skin fibroblasts [12]. Cells were capable of infiltrating the wounds temporised with BTM at a significantly greater rate compared to Integra^®^ both from the wound edge and wound bed (**p* ≤ 0.05). However, we were unable to distinguish graft fibroblasts from infiltrating fibroblasts in allogenic native skin grafts.

### 2.3. Identification of Molecular Mediators that May Influence Graft Take 

In order to shed light on mediators that may influence differences observed in tissue growth, a number of inflammation and wound healing markers were analysed at both RNA and protein levels in grafts. The semi-quantitative real-time PCR analysis (Figure 3) showed an upregulation of some inflammatory markers such as colony-stimulating factor 3/granulocyte colony-stimulating factor (CSF3/G-CSF, *p* < 0.01), chemokine CXC motif ligand 3/macrophage inflammatory protein 2β (CXCL3/MIP-2β, *p* < 0.01) in allogenic native skin and in BTM grafts. Cathepsin G (Ctsg, *p* < 0.01), actin, alpha, cardiac muscle 1 (actc 1, *p* < 0.0001) and integrin alpha 4 (Itga4, *p* < 0.5) were also significantly upregulated in allogenic native skin. Conversely, Integra^®^ grafts had higher expression levels of proliferation phase/anti-inflammatory markers, collagen 3A1 (*p* < 0.01). Cathepsin K (Ctsk), matrix metalloproteinase 2 (MMP2) and matrix metalloproteinase 9 (MMP9) also showed an increased expression trend in Integra^®^ although the differences were not statistically significant. 

At protein level, most inflammatory markers were generally present at higher levels in allogenic native skin and BTM compared to Integra^®^ grafts. Particularly, chemokine CXC motif ligand 13/B lymphocyte chemoattractant (CXCL13/BLC, *p* < 0.05), chemokine CC motif ligand 24 (CCL24/Eotaxin-2, *p* < 0.01), CSF3 (G-CSF, *p* < 0.01), colony stimulating factor 2/granulocyte-macrophage colony stimulating factor (CSF2/GM-CSF, *p* < 0.05), IL-1α (*p* < 0.0001), IL-4 (*p* < 0.01), IL-12p70 (*p* < 0.05), IL-13 (*p* < 0.01), chemokine CXC motif ligand 1/keratinocyte-derived chemokine (CXCL1/KC, *p* < 0.01), chemokine CXC motif ligand 5–6 (CXCL5-6, *p* < 0.001), and chemokine CC motif ligand 9/macrophage inflammatory protein-1 gamma (CCL9/MIP-1γ, *p* < 0.0001) were present at significantly higher concentrations in allogenic native skin, compared to Integra^®^ grafts (Figure 4a–c). Differences between Integra^®^ and BTM grafts in protein levels for individual markers did not reach significance at the two-week time point.

COX-2 is an integral component of inflammation. COX-2 levels, normally present at low levels in the dermis, increases during inflammation in wounds [13]. It has been demonstrated that proinflammatory IL-1α and IL-1β signalling results in COX-2 overexpression [14,15]. To validate the Ptgs2 overexpression observed in BTM grafts, COX-2 protein levels were analysed in grafts using immunohistochemistry (Figure 4d, Appendix A). COX-2 was expressed at least 2-fold higher in fully synthetic BTM grafts compared to collagen-based Integra^®^ grafts (*p* < 0.0001). Conversely, MMP-2 (a marker of remodeling phase) was expressed at higher levels (*p* < 0.0001) in collagen-based Integra^®^ grafts, compared to fully synthetic BTM grafts (Figure 4e, Appendix A). 

### 2.4. Application of Dermal Templates for Bioengineering a Human Skin Equivalent (HSE) 

In addition to the application of dermal templates as dermal grafts, with a need for further skin grafting, we tested whether dermal templates can create a microenvironment for attachment and expansion of human adult skin cells in vitro to bioengineer a definitive cultured skin graft or HSE. We have previously established methodologies to bioengineer HSE with near-native skin architecture using single layer Integra^®^ [16]. In this study, Integra^®^ (single layer) and BTM (single layer) were seeded with adult fibroblast and keratinocytes for construction of HSE (Figure 5a). Both dermal templates were able to support attachment and expansion of adult fibroblasts and keratinocytes in vitro. Two-weeks post grafting immunohistochemical analysis confirmed significantly higher survival of human-derived epidermis and its attachment to the dermis in BTM HSE compared to Integra^®^ HSE grafts (using a human-specific involucrin antibody, **p* ≤ 0.05). Vessels were detected in close proximity to the epidermis, particularly in BTM HSE grafts using CD31 immunohistochemical staining (Figure 5b,d). 

## 3. Discussion

Dermal templates can replace damaged dermis by creating a microenvironment suitable for the host cells to infiltrate and generate neo-dermis. Here, we measured vascularisation and fibroblast invasion as indicative of graft take. We found a significantly more extensive network of vessels and fibroblast infiltration in synthetic BTM grafts, compared to other dermal templates. Protein arrays confirmed the presence of inflammatory chemokines and growth factors two weeks post-surgery in allogenic native skin and BTM grafts. To name some of the most abundant inflammatory markers (Figure 4, reaching over 20% intensity of the summed housekeeping controls) were CXCL5-6, lymphotactin α (XCL-1), CCL1, 2 and 9, and colony-stimulating factor 1/macrophage colony-stimulating factor (CSF1/M-CSF). Based on this collective analysis of the grafts, the authors would like to present a model representing the changes that occur in allogenic native skin and synthetic BTM vs the Collagen-based Integra^®^ grafts microenvironment (Figure 6). In this model, granulocyte and leukocyte subsets (neutrophils, macrophages, mast cell, and lymphocytes) that are likely to sequentially infiltrate the allogenic native skin and synthetic BTM grafts at higher levels are presented. Leukocytes not only serve as immunological effector cells but are also a source of inflammatory and growth-promoting cytokines [17]. Inflammatory interleukins, such as IL-1α, present in these grafts, have been shown to upregulate COX-2, a key mediator of inflammatory response. It has been shown that COX-2 inhibitor reduces CD31 vascularisation of wounds, and COX-2 knockout mice are known to have impaired healing [18,19,20]. Therefore, we postulate that high levels of COX-2 drive vascularisation and fibroblast expansion in allogenic native skin and synthetic BTM grafts.

Proinflammatory macrophages (M1) are known to produce inflammatory cytokines such as IL-1, IL-6, TNF-α and interferon-gamma (INF-γ), whereas, regenerative macrophages (M2) are stimulated with IL-4 and IL-13 and produce a high level of IL-10 and TGF-β [32]. Therefore, it has been postulated that higher IL-10 to IL-6 ratio pushes wounds towards healing. Similarly, a greater ratio of IL-4 or IL-13/ NF-γ would result in enhanced M1 to M2 transition in the macrophage population with known pro-wound healing phenotype [5,33]. As shown in Figure 4, IL-6 expression in collagen-based Integra^®^ grafts is nil, whereas IL-10 is present at low levels in all three types of grafts at the two-week time point. Although low in levels, the higher IL-10/IL-6 ratio in Integra^®^ grafts may explain the decline of COX-2 expression observed in Integra^®^ grafts. 

Collagen dermal templates are resorbed by collagenolytic enzymes. Here, we observed overexpression of a collagenolytic enzyme MMP-2, and up-regulation trend of MMP-9 and Cathepsin K in Integra^®^ grafts. Some of MMP-2 and 9 substrates were collagens, fibronectin, gelatin and laminins [34]. It is the balance between matrix synthesis and proteolytic degradation that drives normal wound healing [34,35]. MMPs ability to degrade ECM proteins, also regulate cell–cell and cell–matrix interaction indirectly through modulating the biological activity and/or releasing of growth factors and cytokines [34]. MMP-2 expression in human is restricted to fibroblasts in scar tissue [36]. It has been shown in full-thickness wound model in mice that MMP-2 and MMP-9 expression increases 10 days post-injury and persists for a number of days [37] which agrees with our findings of MMP-2 overexpression 14 days post-injury.

We hypothesise that a BTM graft induces a greater inflammatory response compared to an Integra^®^ graft. The greater (and prolonged) inflammation response in BTM drives extended proliferation (and tissue growth) phase in both magnitude and length of time, whereas the Integra^®^ grafts have already moved to the remodelling/removal of the inflammatory matrix components phase ahead of BTM grafts. This can be due to the abundance of collagen in Integra^®^ dermal grafts that can modulate fibroblast behaviour. Excessive collagen can trigger fibroblasts to attenuate their growth, and instead, direct them to express ECM proteins and ECM remodelling proteases. This phenomenon may result in dampening the proliferation phase prematurely in Integra^®^ grafts, leading to slower tissue growth when compared to BTM grafts.

It is well accepted in the field that dermal templates must be biocompatible, biodegradable, non-toxic, non-inflammatory, and non-immunogenic. Therefore, when designing dermal templates, there has been a focus to recapitulate the native structure of dermal ECM [38,39]. This study challenges the idea of the need for non-immunogenic and non-inflammatory as requirements for dermal templates and suggests that one should rethink these criteria when developing novel dermal templates. Cell-matrix interaction should be considered when designing novel dermal substitutes. BTM has no known binding site for host cells, and yet it supports more cellular growth than the two collagen matrices tested here. Unlike collagen, which anchors cells via integrins, it is not clear how the host cells attach to BTM, and how this matrix-cell interaction regulates cells behaviour. However, one cannot rule out the slow tissue growth into collagen-based Integra^®^ tested here to be due to presence of glutaraldehyde (commonly used in collagen-based dermal substitutes for cross-linking) that is known to be cell toxic. 

Although skin grafting does not follow classic wound healing stages, this study supports earlier reports that levels of proinflammatory cytokines, such as IL-6 and TNFα, are drastically reduced in mouse wounds that are healed [40]. Prolonged inflammation has been associated with hypertrophic scarring and fibrosis in patients [7], but whether the higher (magnitude and length) inflammation in BTM grafts compared to Integra^®^ grafts would have a negative effect on scarring in these grafts requires further investigation. Wound contraction also plays a role in fibrosis, and although some contraction was observed in Integra^®^ (Appendix A), this, may not be a true indication of long term wound contraction in patients. At the two-week time point, BTM and Integra^®^ grafts still retained their relatively inflexible seals which resist contraction. Longer time points (post seal removal) studies are needed to compare templates for clinically relevant wound contraction. 

Spatiotemporal cues in the wound bed are constantly evolving, and additional time points are required to elucidate a better understanding of cell-matrix interaction and their long-term consequences in skin grafting. Nevertheless, this study has identified some key differences between a collagen-derived and a fully synthetic polyurethane dermal graft, which should be considered when selecting for clinical application.

## 4. Materials and Methods

### 4.1. Mouse Skin Grafting

The protocol and procedures were ethically reviewed and approved by the Alfred Research Alliance Animal Ethics Committee (approval E/1665/2016/M, 04/08/2016) and followed the Declaration of Helsinki Principles. Skin grafting was performed as described previously [41]. Briefly, male athymic nude mice aged 10–12 weeks were anaesthetised with isoflurane (2 L/min) and a full-thickness surgical wound created by excising a circular section of skin 1.2 cm in diameter on their dorsal side approximately 1 cm below the occipital protuberance and several millimetres to the left of the midline. Mice were grafted with bilayered polyurethane-sealed NovoSorb^®^ BTM (PolyNovo, Australia); bilayered silicone-sealed Integra^®^ (Life Sciences Corp., Plainsboro, NJ, USA); full-thickness skin harvested from C57BL/6 mice, or HSE. The wound was dressed and sealed with SurfaSoft^®^ (Taurenon, The Netherlands), Tegaderm (3M, St. Paul, MN, USA) and Coban™(3M, Australia) for the duration of the experiment. Mice were killed, and grafts were analysed after two weeks.

### 4.2. Wound Healing Markers and ECM Expression Levels by Real Time PCR

Tissues (*n* = 3 mice per group) were homogenised with an IKA Ultra-Turrax T-25 disperser (Janke and Kunkel, Germany) and processed for RNA extraction using an RNeasy mini kit (Qiagen, Germany) according to manufacturer instructions. Eluted RNA received 10 U RNasin Plus RNase inhibitor (Promega, Madison, WI, USA) and was quantified on Quantus Fluorometer using QuantiFluor RNA system kit (Promega, Madison, WI, USA). 

Mouse Wound Healing RT^2^ PCR Profiler arrays (Qiagen, Germany) were performed according to the manufacturer instructions. Briefly, 400 ng of RNA template was added to genomic DNA elimination mix (RT2 First Strand kit, Qiagen, Germany) in a total volume of 10 µL and incubated 5 min at 42 °C, followed by 1-minute incubation on ice. Reverse transcription mixture was added to a total volume of 20 µL and the mix was incubated for 15 min at 42 °C and 5 min at 95 °C, diluted by adding 91 µL RNase-free water and frozen at −30 °C. cDNA product was mixed with SYBR Green Mastermix, RNase-free water and amplified using LightCycler 480 (Roche, Indianapolis, IN, USA). Graft gene targets with >2-fold change, as compared to host, were converted to log_2_ base and a heatmap plotted (Heatmapper online platform). 

### 4.3. Antibody Array 

Mouse inflammation antibody arrays (Abcam, UK) were performed according to the manufacturer’s instructions (*n* = 4 per group). Briefly, 100 ug of mouse graft protein was added to blocked array membrane for overnight incubation at 4 °C on a rocking platform. Membranes received 40 min wash in 20 mL of Buffer I, followed by three 5 min washes in Buffer I, and two 5 min washes in Buffer II. Membranes were incubated 1.5 hours while rocking at R/T with biotinylated anti-cytokine antibodies. This was followed by washes in Buffer I and II and incubation with streptavidin horseradish peroxidase conjugate for 2 hours. Signal was developed using chemiluminescent substrate and signals collected from each target on the membrane using ChemiDoc imaging system (Bio-Rad, Hercules, CA, USA). Signal density was assessed with Image J software protein array analyser plugin (NIH), collected as an integrated area for each target, present in duplicate on each membrane. Averaged signal per target was modified with negative control and blank signals, followed by normalisation to positive controls from the same blot. 

### 4.4. Immunohistochemistry and Analysis

Graft vascularisation (*n* = 4 mice per group) was assessed with CD31 staining, as described previously [41]. Briefly, cryopreserved sections were fixed in 2% formalin (20 min) and permeabilised in 80% methanol at −20 °C (15 mins) followed by 3 × 5 min PBS washes. Endogenous peroxidase activity was quenched by incubating the sections with 3% hydrogen peroxide (H_2_O_2_) for 15 mins at room temperature, followed by 3 × 5 min PBS washes. Sections were blocked in 10% normal goat serum (Applied Biological Products Management, Australia) in 10% bovine serum albumin (MP Biomedicals, Australia) in PBS for 2 h at room temperature, followed by incubation in rat anti-CD31 antibody (1:100; BD Biosciences, San Jose, CA, USA) overnight at 4 °C. Slides were incubated in a biotinylated goat anti-rat antibody (1:500, BD Biosciences, San Jose, CA, USA) for 45 mins at room temperature, followed by 3 × 5 min PBS washes. After incubation with avidin-biotin complex for 30 min (Vector Laboratories, Burlingame, CA, USA), the colour was developed with diaminobenzidine/H_2_O_2_ (Vector Laboratories) followed by counterstaining with haematoxylin. Four fields of view scanning the whole depth of the graft from each section stained with CD31 antibody were imaged at ×40. Image analysis was performed using Image J software ( ) using the colour threshold method. The threshold value was kept constant across all sections. Blood vessel area, marked by CD31^+^ endothelial cells, was normalised for image area. Average vessel diameter was measured by taking three measurements per each stretch of the vessel in the field of view in each image [42]. Human involucrin was detected in HSE grafts following involucrin immuno-kit instructions (Biomedical Technologies, Stoughton, MA, USA). Briefly, formalin-fixed, paraffin-embedded sections were deparaffinised and endogenous peroxidase activity quenched with 0.3% H_2_O_2_ in tris-saline pH 7.6 for 30 minutes. After washes in tris-saline, sections were blocked in normal goat serum overnight at 4 °C. The blocking solution was replaced with rabbit anti-human involucrin solution for 1 hour at room temperature (t°C). Sections were washed in tris-saline again and incubated with goat anti-rabbit Ig for 30 mins. Final washes were followed with diaminobenzidine (DAB)/H_2_O_2_ (Vector Laboratories, Burlingame, CA, USA) colour development and haematoxylin counterstaining. For MMP-2 and COX-2 detection, sections were antigen retrieved in citrate buffer pH 6.0. Sections were incubated with rabbit anti-MMP-2 antibody (Abcam, UK) at 1:200 or rabbit anti-COX-2 antibody (Abcam, UK) at 1:2000, followed by goat anti-rabbit Ig-HRP conjugate (Cell Signalling, Danvers, MA, USA) incubation. Similar to involucrin detection, DAB developed sections were counterstained. All IHC stained images were analysed on FIJI software for positive cellular area expressed as a % of the full image area. The threshold value was kept constant across all sections.

### 4.5. Confocal Microscopy

Confocal microscopy was performed as described previously with minor modifications [43]. Briefly, cryopreserved graft sections (*n* = 4–6 mice per group) were permeabilised with methanol (10 min), washed, blocked (30 min), and incubated with rabbit anti-vimentin antibody (1:600, Cell Signalling, Danvers, MA, USA) overnight. Sections were washed and probed with donkey anti-rabbit Ig–AF647 antibody (Invitrogen, Carlsbad, CA, USA) for 60 min and washed excessively prior to mounting using Prolong Gold (Invitrogen, Carlsbad, CA, USA). For analysis, high-resolution images were acquired on Nikon A1R point scanning confocal microscope with Plan Fluo 20 x MIm/0.75 NA objective × 2 optical zoom. Six fields of view were randomly selected in specified areas of the graft. Integra^®^ spectral profile in the 405–550 nm range interfered with fibroblast detection and counting. We, therefore, performed spectral unmixing of the stained Integra^®^ graft images using NIS Analysis (Nikon, Japan), assigning all Integra^®^ spectral property to the green channel (500–550nm). Evaluation of fibroblast infiltration was performed by thresholding vimentin-positive cells, counted with NIS Analysis software (Nikon, Japan).

### 4.6. Scanning Electron Microscopy 

FEI Nova NanoSEM 450 FEGSEM scanning electron microscope was used to collect high-resolution secondary electron images from the cross-section of dermal templates. The microscope was operated at field-free mode, 3kV accelerating voltage, spot size 3.0 and aperture size 30 µm. Prior to cross-sectioning Integra^®^ was fixed using 10% normal buffered formalin and air-dried). Native mouse skin was fixed in 2.5% glutaraldehyde and 2% paraformaldehyde in 0.1M sodium cacodylate buffer overnight at 4 °C. Then it was washed three times in fresh sodium cacodylate buffer, before being postfixed in 1% osmium tetroxide and 1.5% potassium ferricyanide in cacodylate buffer for 2 hours at RT. Post-fixation washed sample five times with Milli-Q water for 30 mins each step. The tissues were dehydrated in increasing concentrations of ethanol, consisting of 30, 50, 70, 90 and 2 × 100% ethanol for 60 minutes each step. Dehydrated tissues were dried with a Bal-Tec CPD 030 critical point dried and mounted onto 12 mm diameter aluminium SEM stubs using sticky carbon tabs. All samples were cross-sectioned, mounted on aluminium stubs using carbon tape and sputter-coated with iridium using a Cressington 208 HR sputter coater.

### 4.7. Human Skin Equivalent (HSE)

HSE was constructed according to our previous study with some modification [16]. Briefly, adult fibroblasts and keratinocytes were isolated from discarded skin (under patients’ consent and approved by the Monash Ethics Committee). Fibroblasts were expanded in single layer dermal templates, Integra^®^ and BTM (with no sealing film), in DMEM with bovine calf serum (10%, Sigma, St. Louis, MI, USA) and gentamicin (50 µg/mL, Life Technologies, Carlsbad, CA, USA) 4–7 days. Dermal templates were soaked in human plasma (20–25 mg/mL) and CaCl_2_ (1%) for 30 min at 37 °C prior to seeding keratinocytes. The HSEs were allowed to expand and stratify in Green’s media: DMEM: F12 media (3:1) (Life Technologies, Carlsbad, CA, USA) supplemented with L-glutamine (4 mM, Life Technologies, Carlsbad, CA, USA), adenine (0.18 mM, Calbiochem, San Diego, CA, USA), hydrocortisone (0.4 ug/mL, Calbiochem, San Diego, CA, USA), triiodothyronine (T3)( 2 × 10^−9^ M, Sigma), insulin (5 ug/mL, Sigma, St. Louis, MI, USA), transferrin (TRF) (5 ug/mL, Sigma, St. Louis, MI, USA), Epidermal Growth Factor (EGF) (10 ng/mL, R&D Systems, Minneapolis, MI, USA), foetal calf serum (10%, Thermo Fisher Scientific, Waltham, MA, USA), and gentamicin (50 µg/mL, Life Technologies, Carlsbad, CA, USA) 10 to 14 days prior to grafting (*n* = 5–6 mice per group). 

## Figures and Tables

**Figure 1 ijms-21-04508-f001:**
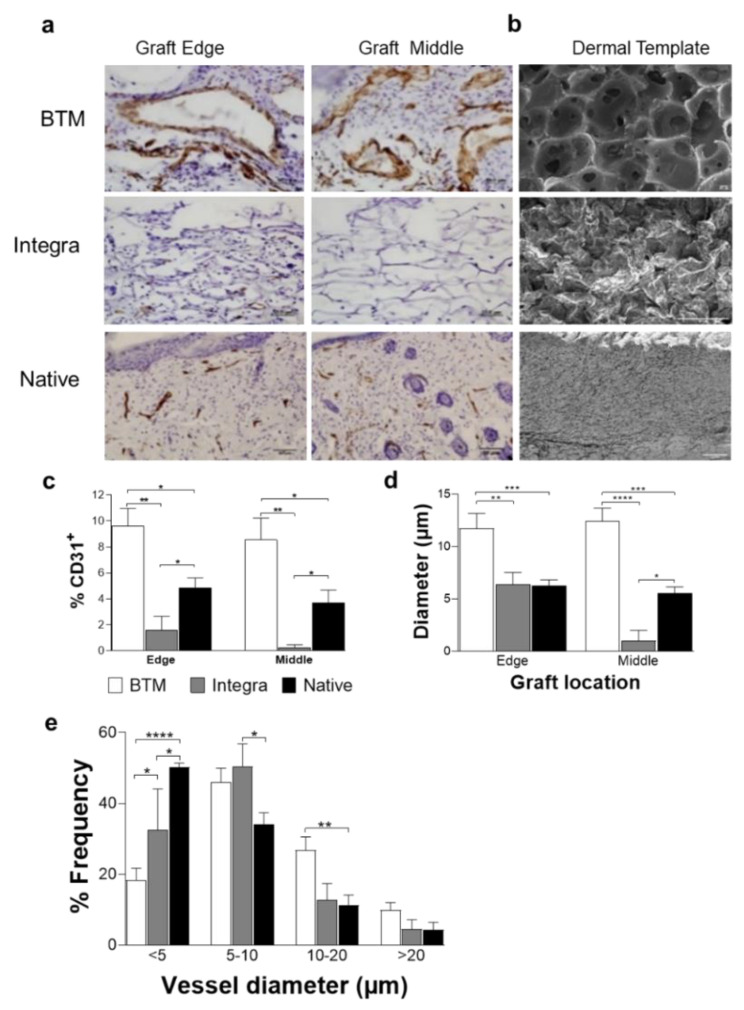
Detection of endothelial cells in mouse grafts. (**a**) CD31 positive endothelial cells were detected in grafts using IHC staining. Representative images of BTM, Integra^®^, and allogenic native skin grafts are presented (scale bar 50 µm). (**b**) SEM images of BTM, Integra^®^, and native mouse skin (scale bar 100 µm). (**c**) CD31 staining on graft edge and the middle was quantified and normalised for image area. CD31 staining was used to score vessel diameter (**d**), and the frequency of different vessel diameters (**e**). Values represent mean +/- SEM in each group (*n* = 4 mice per group) and analysed using unpaired t-test. * = *p* ≤ 0.05, ** = *p* ≤ 0.01, *** = *p* ≤ 0.001, **** = *p* ≤ 0.0001.

**Figure 2 ijms-21-04508-f002:**
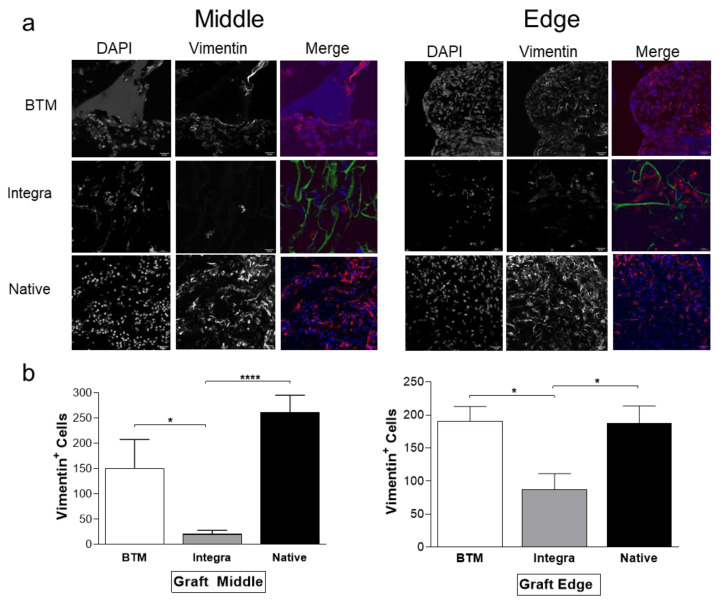
Mesenchymal host infiltration in mouse grafts. (**a**) Fibroblasts were detected by confocal microscopy using a vimentin antibody (in red) and DAPI (in blue). Dermal template autofluorescence is shown in green (scale bar 50 µm). (**b**) Vimentin positive cells were quantified on NIS Analysis software (Nikon, Japan) in six fields of view (*n* = 4–6 mice per group). Values represent mean +/− SEM in each group and analysed using unpaired t-test. * = *p* ≤ 0.05, **** = *p* < 0.0001.

**Figure 3 ijms-21-04508-f003:**
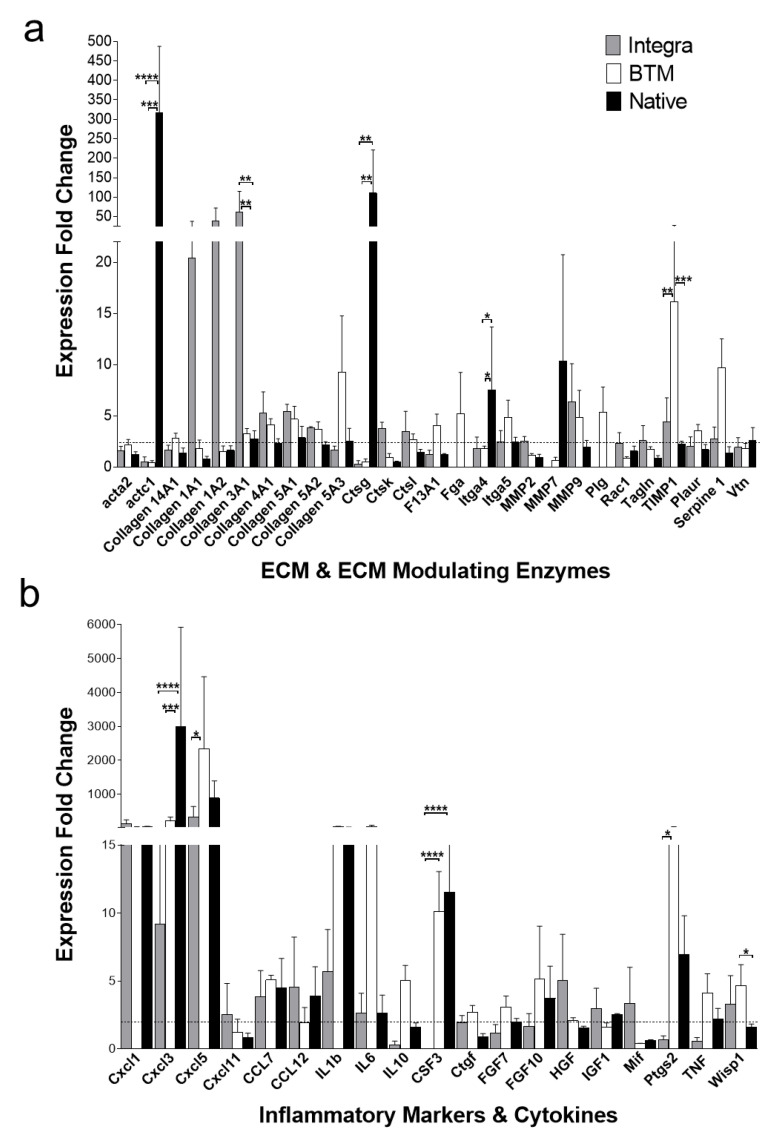
RNA expression profiling (mouse wound healing RT^2^ profiler PCR array) of 2-week-long grafts. (**a**) Upregulated ECM structural and modifying enzyme genes identified in grafts by C_t_ comparison with host RNA. (**b**) Upregulated inflammation markers, including cytokines and chemokines. Selected targets produced an average of >2-fold change in at least one of the studied groups. Mean and SEM values presented for each group (*n* = 3 per group). * = *p* < 0.05, ** = *p* < 0.01, *** = *p* < 0.001, **** = *p* < 0.0001). A heatmap of all arrayed genes is available in Appendix A.

**Figure 4 ijms-21-04508-f004:**
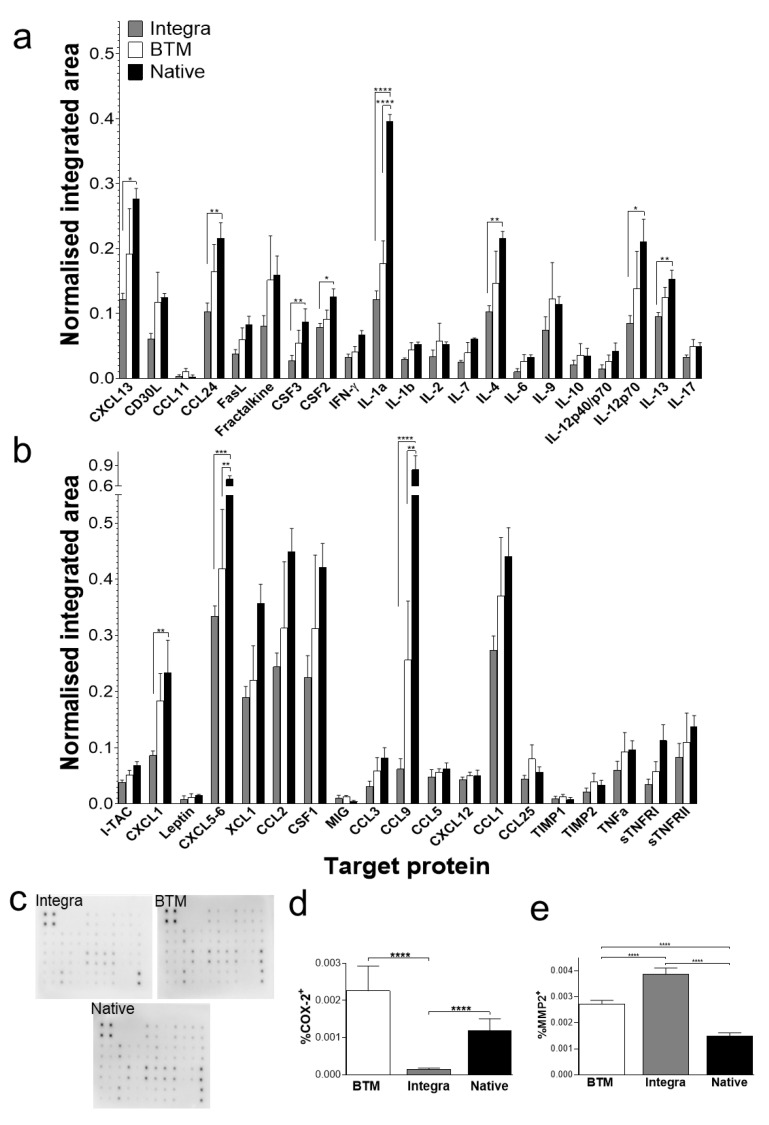
Protein expression profiling of 2-week-long grafts. (**a**,**b**) Protein signals recorded for each target using 10-second exposure and normalised to positive controls. Mean and SEM values presented for each target (*n* = 4 per group) (**a**) Targets from rows 1 and 2 position F6 to rows 5 and 6 position B2 of the array). (**b**) Targets from rows 5 and 6 position B3 to rows 7 and 8 position I9 of the array. The full map of the array is provided in Appendix A. (**c**) Representative chemiluminescent protein array blots for each group. (**d**) Quantification of immunoperoxidase staining for inflammation marker, COX-2 (*n* = 4–8 mice per group), using unpaired t-test * = *p* < 0.05, ** = *p* < 0.01, *** = *p* < 0.001, **** = *p* < 0.0001. (**e**) Quantification of immunoperoxidase staining for ECM remodelling enzyme, MMP-2 (*n* = 4–8 mice per group). Results analysed using unpaired t-test, significant *p*-values as for panel c. Abbreviations: CCL1 (TCA-3), chemokine CC motif ligand 1; CCL2 (MCP-1), chemokine CC motif ligand 2; CCL3 (MIP-1α), chemokine CC motif ligand 3; CCL5 (RANTES), chemokine CC motif ligand 5; CCL9 (MIP1γ), chemokine CC motif ligand 9; CCL11 (Eotaxin), chemokine CC motif ligand 11; CCL24 (Eotaxin-2), chemokine CC motif ligand 24; CCL25 (TECK), chemokine CC motif ligand 25; CSF1 (M-CSF), colony-stimulating factor 1; CSF2 (GM-CSF), colony stimulating factor 2; CSF3 (G-CSF), colony stimulating factor 3; CXCL1 (KC), chemokine CXC motif ligand 1; CXCL5-6 (LIX), chemokine CXC motif ligand 5-6; CXCL12 (SDF-1), chemokine CXC motif ligand 12; CXCL13 (BLC), chemokine CXC motif ligand 13; IL1 or 6 or 10, interleukin 1 or 6 or 10; TIMP, tissue inhibitors of metalloproteinases; TNF, tumour necrosis factor, XCL1, lymphotactin α.

**Figure 5 ijms-21-04508-f005:**
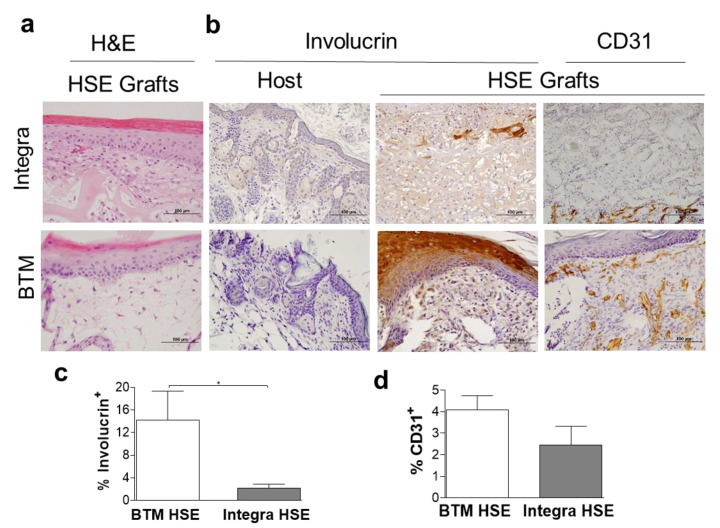
Neo-epidermis detection in 2-week-long HSE grafts. (**a**) Representative H&E staining of HSE constructed using Integra^®^ (single layer) or BTM (single layer) two weeks post grafting (scale bar 100 µm). (**b**) Representative images of human-specific involucrin staining of mouse host and HSE grafts and CD31 in vivo (scale bar 100 µm). (**c**) Human involucrin staining of HSE grafts was quantified and Mean and SEM values are presented (*n* = 5–6 per group, **p* value < 0.05). (**d**) CD31 expression was also quantified in HSE grafts. Mean and SEM values are presented (*n* = 5–6 per group).

**Figure 6 ijms-21-04508-f006:**
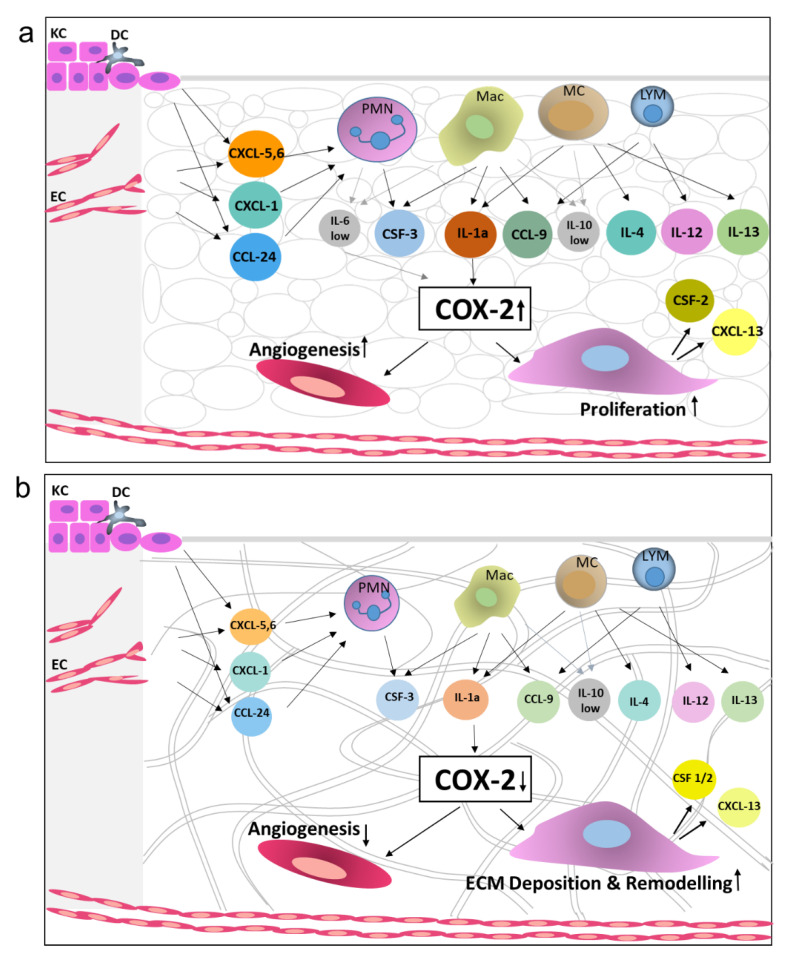
A schematic diagram representing the key differences between Integra, BTM and allogenic native skin grafts in terms of inflammatory and wound healing mediators’ expression levels. Growth factors showing differential expression at protein levels in grafts and cell types known to secrete these growth factors are presented. (**a**) Allogenic native skin graft (and to a lesser degree BTM graft) triggers significantly higher inflammation response in the host, compared to (**b**) Integra^®^, resulting in greater fibroblast growth and vascularisation. Integra^®^ grafts, however, are more progressed towards ECM remodelling phase within this time point, possibly triggered by an excessive amount of collagen in the graft. The arrows suggest pathways, based on the literature below, which link the growth factors to the grafting outcomes. Only the feed-forward pathways (and no inhibitory pathways) are presented. Abbreviations: polymorphonuclear cells (PMN), macrophages (Mac), mast cells (MC), basal keratinocytes (KC), endothelial cells (EC), and dendritic cells (DC) [17,19,21,22,23,24,25,26,27,28,29,30,31].

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
