# Peer review of "A Comparative Study of Engineered Dermal Templates for Skin Wound Repair in a Mouse Model"

_ijms, 2020, doi:10.3390/ijms21124508_

Round 1

Reviewer 1 Report

This study on dermal templates for wound healing is very interesting and highlights some important comparisons and results. It is well written and comprehensive.

I would just highlight a couple of points:

  1. Page 2, line 65 and throughout the manuscript: Please add full names of each gene or protein before abbreviating.
  2. Materials and methods: How many times was each experiment performed? Please quote n numbers after each description. This is especially important in PCR arrays which should be carried out a minimum of 3 times for the results to be reliable.

Author Response

Dear reviewer,

Thank you for the complementary feedback.

  • The full name of each gene / protein has been added to abstract on page 1 line 25; page 2 lines 65-68; page 5 lines 129-135; page 6 lines 147 &148; page 7 lines 149-153; page 9 lines 213 and 214; and page 10 line 239 of the manuscript as requested.
  • The methods have been improved. The number of times each experiment has been performed was stated in each figure legend. This number is now added to each section in M&M as requested by the reviewer on page 12 lines 306 and 320 and 332; page 13 line 363; and page 14 line 400.

Please find attached details track changed in “red and underlined” throughout the manuscript.

Reviewer 2 Report

The MS is about the comparison of engineered dermal templates for skin wound repair. It is well organized and written. In my point of view it can be accepted in its present form.

Author Response

Dear reviewer,

Thank you for the complementary feedback and accepting the manuscript for publication.  Please find attached the revised version of the manuscript for your information. Changes are in "red and underlined". 

Kind Regards,

Shiva Akbarzadeh

Reviewer 3 Report

Banakh et al present an interesting and well-designed investigation of the modulation of biochemical balance between fibrosis and neovascularization in the setting of artificial skin grafts to full thickness injuries in a nude mouse model.  The authors also include assessments of human fibroblast and keratinocyte within the artificial matrixes that were used “off the shelf”.

Multiple modalities of assessment were made that included antibody arrays, immunohistochemistry, confocal microscopy and scanning electron microscopy.

As illustrated in the authors figure 6, the difference in the balance of inflammation or wound healing based on multiple modulators could be simplified to changes in COX-2 upregulation.

Overall, the work is significant and well-supported by the data provide to the extent the readership can interpret it.

Unfortunately, as outlined subsequently, the data presentation has poor resolution, with complicated figures that cannot be read and interpreted.  If overcome, this will be a fine paper.

Figure 1 legend is almost not able to be read without a magnifying glass.  Panels C-E of this figure are also not easily assessed.  The authors must resize these panels of figure 1 and perhaps split figure 1 into two separate illustrations.

Figure 2 and its legend are also too small to assess properly.

Lines 112-114.  The two sentences must be combined somehow.

Figure 3 is worse than figures 1 and 2 – Panels A and B have statistical comparisons and X axis text I cannot read.  The C panel is complex, with small and poorly focused/fuzzy side text that cannot be read.

Figure 4 is nearly unreadable.

Figure 5 is the only reasonably interpretable figure.

Figure 6, the critical synthesis figure, again severely suffers from resolution issues and poor readability.

Author Response

Dear Reviewer,

Thank you for the complementary feedback.

  • All figures quality and resolution have been optimised.
  • Figure 3 (page 6, line 139) has been broken up as suggested by the reviewer and the heat map has been moved to supplementary data S2.
  • Lines 112-114 in the previous version ares now corrected. Please see lines 134 – 136 of the revised version.

Please see attached a copy of the revised manuscript. 

Kind Regards,

Shiva Akbarzadeh

Reviewer 4 Report

Author tried to investigated the wound bed and acellular “off the shelf” dermal template interaction in a mouse model. Full thickness wounds in nude mice were grafted with allogenic skin, and either collagen-based, or fully synthetic dermal templates, described a promising template, based mainly on the vascularization and fibroblast infiltration and making human skin equivalent or HSE. Today, engineered skin grafting has made great progress, authors focused on the which kind of dermal templates is better on hosting cells and tried to illuminate the mechanism. I pick up partial points to make your attention as followings. I suggest authors to arrange the reasonable and logic story comprehensively.

  1. The resolution of the figures and figure legends is low, when enlarge the pictures, I can’t see the labels.
  2. There is no illumination that why the mesenchymal host infiltration in different places (middle & edge) of mouse grafts is different, especially on integra template.
  3. You chose CD31& Vimentin to mark hosted endothelial cells & mesenchymal cells, I think only one marker is not representative, and it is not rigorous to show the positive rate by quantify positive cells in different image areas.
  4. The introduction of COX-2 was farfetched, there is no experiment result about its function.
  5. The structures of HSE Grafts are not perfect, there is a gap between epidermis and dermis.

Author Response

Dear Reviewer,

Thank you for the complementary feedback.

  • All the figures and figure legends have been resized and resolutions optimised according to the reviewer’s instructions. The heat map in Figure 3 has been moved to supplementary data S2.
  • As recommended by the reviewer the explanation has been given as to why the grafts were analysed in the middle and the edge on page 3 line 90: “The middle of the grafts was analysed separately from the graft edge as an indication of the endothelial cell infiltration from the wound bed. It is important to measure the infiltration of dermal templates from the wound bed in a mouse model, as the wound edge contribution in healing large surface area grafts in patients would be relatively small.”
  • Representative images of H&E staining of the BTM, Integra and native skin grafts have been added to the results on page 3 line 88 and supplementary data S1 for additional support of cell infiltration, showing the abundance of red blood cells in BTM grafts, compared to Integra grafts. To quantify vascularisation, grafts were analysed not only by quantifying CD31 distribution, but also by measuring the diameter of all the vessels in three locations perpendicular to the stretches of vessels in grafts using CD31 IHC images. The entire depth of dermis on the edge and the middle of each graft was analysed. Details has been added to M&M page 12, lines 333 - 345. Vimentin is a well-studied and widely used marker of skin fibroblasts. It is expressed by all fibroblasts subtypes and therefore appropriate to use when reporting overall fibroblast infiltration. This has been added to page 4 line 115.
  • Ptgs 2 (COX-2) RNA levels was initially found to be high in BTM grafts, which prompted us to measure its protein levels in grafts. The rational and relevant references have been added to page 8, lines 175 - 177 as requested by the reviewer.
  • The images of the HSE in Figure 5 have been replaced to show correct structure of dermis / epidermis as requested by the reviewer. The Figure 5 result has also been updated in page 8, lines 194 - 197.

Please find attached the revised version of the manuscript including the supplementary data.

Kind Regards,

Shiva Akbarzadeh 

Round 2

Reviewer 3 Report

A great improvement, but if there is a way, figure 4b needs to perhaps be divided in two somehow.  There are far too many x axis entries that are compressed to allow one to see what subcondition is different from the others with ease.

Author Response

Thank you for reviewing the manuscript. Your concern has been addressed and track changed in red and underlined in the manuscript. Specifically, Figure 4b has been split into Figure 4a and 4b and spacing between each protein marker has been increased. The figure 4 legend on page 8, lines 158 – 175 is updated.
